# Thiadiazolidinone (TDZD) Analogs Inhibit Aggregation-Mediated Pathology in Diverse Neurodegeneration Models, and Extend *C. elegans* Life- and Healthspan

**DOI:** 10.3390/ph16101498

**Published:** 2023-10-20

**Authors:** Samuel Kakraba, Srinivas Ayyadevara, Nirjal Mainali, Meenakshisundaram Balasubramaniam, Suresh Bowroju, Narsimha Reddy Penthala, Ramani Atluri, Steven W. Barger, Sue T. Griffin, Peter A. Crooks, Robert J. Shmookler Reis

**Affiliations:** 1Department of Geriatrics, University of Arkansas for Medical Sciences, Little Rock, AR 72205, USA; nmainali@uams.edu (N.M.); mbalasubramaniam@uams.edu (M.B.); ratluri@uams.edu (R.A.); bargerstevenw@uams.edu (S.W.B.); griffinsuet@uams.edu (S.T.G.); 2Central Arkansas Veterans Healthcare Service, Little Rock, AR 72205, USA; 3Department of Pharmaceutical Sciences, University of Arkansas for Medical Sciences, Little Rock, AR 72205, USA; skbowroju@gmail.com (S.B.); nrpenthala@uams.edu (N.R.P.); pacrooks@uams.edu (P.A.C.)

**Keywords:** aggregation, Alzheimer’s disease, glycogen synthase kinase 3β (GSK3β), molecular modeling, neurodegeneration, neurodegenerative disease, protein aggregation, proteostasis, thiadiazolidinones (TDZDs)

## Abstract

Chronic, low-grade inflammation has been implicated in aging and age-dependent conditions, including Alzheimer’s disease, cardiomyopathy, and cancer. One of the age-associated processes underlying chronic inflammation is protein aggregation, which is implicated in neuroinflammation and a broad spectrum of neurodegenerative diseases such as Alzheimer’s, Huntington’s, and Parkinson’s diseases. We screened a panel of bioactive thiadiazolidinones (TDZDs) from our in-house library for rescue of protein aggregation in human-cell and *C. elegans* models of neurodegeneration. Among the tested TDZD analogs, PNR886 and PNR962 were most effective, significantly reducing both the number and intensity of Alzheimer-like tau and amyloid aggregates in human cell-culture models of pathogenic aggregation. A *C. elegans* strain expressing human Aβ_1–42_ in muscle, leading to AD-like amyloidopathy, developed fewer and smaller aggregates after PNR886 or PNR962 treatment. Moreover, age-progressive paralysis was reduced 90% by PNR886 and 75% by PNR962, and “healthspan” (the median duration of spontaneous motility) was extended 29% and 62%, respectively. These TDZD analogs also extended wild-type *C. elegans* lifespan by 15–30% (*p* < 0.001), placing them among the most effective life-extension drugs. Because the lead drug in this family, TDZD-8, inhibits GSK3β, we used molecular-dynamic tools to assess whether these analogs may also target GSK3β. In silico modeling predicted that PNR886 or PNR962 would bind to the same allosteric pocket of inactive GSK3β as TDZD-8, employing the same pharmacophore but attaching with greater avidity. PNR886 and PNR962 are thus compelling candidate drugs for treatment of tau- and amyloid-associated neurodegenerative diseases such as AD, potentially also reducing all-cause mortality.

## 1. Introduction

Many neurodegenerative diseases have increased in incidence over the past several decades, including Alzheimer’s disease (AD), Huntington’s disease (HD), and Parkinson’s disease (PD) [1,2,3] in part due to the demographic shift to a more aged population [4,5,6]. Three common features linking these pathologies are their age-progressive nature, association with chronic inflammation [7], and cytotoxic protein aggregation, a biological phenomenon wherein misfolded proteins coalesce in intra- or extracellular conglomerates [8,9,10,11,12,13,14,15,16,17,18,19].

AD impairs cognition and memory of humans, a property also apparent in animal models with pathology mimicking AD. Genetic factors (such as ApoE polymorphism), and non-genetic factors including age and diet, have been implicated in the relative risk and etiology of AD [3,20,21,22,23]. Among the factors predisposing to development of AD, chronic inflammation is noteworthy as a distinctive and ubiquitous feature of normal aging [7,24,25]. Pharmacological interventions, including certain nonsteroidal anti-inflammatory drugs (NSAIDs) such as cholinesterase inhibitors (e.g., donepezil, galantamine, and rivastigmine) among other drugs, have shown limited protection against AD, PD, and related dementias in prospective trials—but appear unable to reverse associated damage once it has occurred [26,27,28,29,30,31]. Although several non-pharmacological interventions (e.g., intake of dietary supplements including polyunsaturated ω-3 fatty acids) have been suggested to delay or reduce the severity of AD, further studies indicated that the protective benefits of such supplements were confounded by other dietary factors, the stage of AD, and the apolipoprotein E genotype [32,33,34,35,36]. Currently, no strategies or approved medications have been demonstrated to attenuate or reverse AD dementia once it has been diagnosed.

Thiadiazolidinones (TDZDs), small heterocyclic compounds derived from thiadiazolidine, include a wide range of molecules with potent anti-inflammatory properties [37]. TDZD-8 (2-methyl-4-(phenylmethyl)-1,2,4-thiadiazolidine-3,5-dione) is the best-studied TDZD and was reported to be a non-ATP-competitive inhibitor of glycogen synthase kinase 3β (GSK3β), a serine/threonine kinase [38]. In a recent study, we presented structural evidence that TDZD-8 binds the hydrophobic or allosteric pocket of GSK3β in its *inactive* conformation and impedes its return to the active state [39]. This property, and the very low enzyme-turnover rate observed, have significant implications for the potential of TDZD-8 as a deterrent for age-associated diseases such as AD [40,41].

We had previously shown that several anti-inflammatory compounds, including a combrestastatin analog (PNR502), aspirin, and several quinoline analogs, inhibit protein aggregation and aggregation-mediated pathology in human cell-culture and intact-animal models of neurodegenerative diseases [18,42,43]. We demonstrate in the current work that two novel members of the TDZD family, PNR886 and PNR962, were more potent than TDZD-8 in delaying and attenuating AD-like protein aggregation; this applied to both human-cell and *C. elegans* models of tauopathy and Aβ_1–42_-driven amyloidopathy. In AD-model and wild-type *C. elegans*, lifespan and healthspan were substantially extended by both PNR886 and PNR962, suggesting that their benefits may also alleviate other age-progressive declines.

## 2. Results

We utilized an in-house library of novel TDZD analogs, synthesized through standard medicinal chemistry methods reported previously [37,44], and tested them for efficacy to inhibit aggregation. Two of the fifty-five TDZD analogs tested, PNR886 and PNR962 (see structures in Figure 1), demonstrated superior pharmacological benefits to oppose aggregate accumulation in human cell-culture models of AD-like tauopathy and amyloidopathy.

The primary drug screen employed HEK293-tau cells (also called HEK-tau), adenovirus-transformed human embryonal kidney cells expressing a transgene that encodes normal tau. HEK293 cells express many transcripts typical of neurons, including those encoding neurofilament and ion-channel proteins, neurogranin, internexin, vimentin, synapse-associated proteins, and neuron-specific forms of hexokinase and enolase [45]. Amyloid-like aggregates, stainable with thioflavin T, accrue in HEK-tau cells but are markedly reduced by exposure to TDZD-8 or several related compounds. Two analogs, PNR886 and PNR962, were significantly more potent than TDZD-8 in opposing aggregation, by >60-fold (Figure 2), while displaying no detectable cytotoxicity in these cells at doses up to 10 µM. At 1 mM, all drugs exhibited 70–90% cytotoxicity. The same two analogs conferred similar protection to SH-SY5Y-APP_Sw_ neuroblastoma cells, which express a transgene encoding APP_Sw_, a “Swedish” double mutant of amyloid precursor protein observed in familial-AD pedigrees. Unlike HEK-tau, these cells showed mildly adverse effects of these drugs, as evidenced by ~17% decline in cell numbers at 1 µM. Both PNR962 and PNR886 inhibited GSK3β kinase activity in vitro with IC_50_ values more than 10-fold lower than TDZD-8.

### 2.1. PNR886 and PNR962 Reduce Aggregation in C. elegans Models of Neurodegeneration

We tested these TDZD analogs at a range of doses in several C. elegans models of neurodegeneration-associated protein aggregation. Both PNR886 and PNR962 rescued age-dependent accumulation of punctate aggregates in C. elegans strain AM141, expressing polyglutamine fused in-frame to yellow fluorescent protein (Q40::YFP), a nematode model of Huntington’s disease (Figure 3A,B,D), over a range of doses—only the best of which are shown in Figure 3. PNR886 was roughly twice as effective as PNR962 in reducing the number of aggregates (Figure 3A), whereas PNR962 appeared to be more effective in reducing the mean total aggregate fluorescence per worm (Figure 3B), presumably by limiting aggregate size and/or Q40::YFP content per aggregate (Figure 3).

We next assessed whether PNR886 and PNR962 protect *C. elegans* from paralysis arising from aggregation, in *C. elegans* strain CL4176 (expressing human Aβ_1–42_ in body-wall muscle). Young adults accumulate amyloid plaque and undergo acute paralysis within two days after induction of Aβ_1–42_ synthesis in late larvae, just preceding the adult stage. Treatment with 10 µM PNR886 or PNR962, beginning at the L4 larval induction, reduced paralysis by >90% and >75%, respectively (Figure 3C).

### 2.2. TDZD Analog Treatments Extend Lifespan and Healthspan of Wild-Type Nematodes

Considering that many interventions that attenuate protein aggregation also extend lifespan [46,47,48,49], we asked whether exposure of wildtype C. elegans (wild-type strain Bristol-N2, of the longest-lived DRM lineage) to these TDZD analogs, beginning at egg hatching, would improve their survival. Adult worms were lysed and ~60 eggs placed on each drug or control plate, as described previously [18]. Groups of worms (*N* ≈ 50) were picked at random on day 3.5 of adult life and transferred to fresh plates with the same drug or vehicle. Live *C. elegans* worms were subsequently transferred to fresh plates on alternate days, noting the numbers of worms alive, or dead by “natural causes”, until no worms remained. Worms lost due to causes other than natural death were censored at the midpoint between “alive” and “lost/dead” observations. Compared to the mock treatment (vehicle without a drug), 10 µM PNR962 or PNR886 extended the mean lifespan of wild-type nematodes by ~15%; whereas 100 µM PNR886 extended it by ~30% (Figure 4; each *p* < 0.001). Addition of PNR962 at 100 µM caused developmental delay and therefore was not continued.

To assess “healthspan”, we employed an assay that we had developed previously [50] to monitor the age-dependent onset of paralysis in CL4176 adults carrying a transgene for muscle expression of Aβ_1–42_. In contrast to the previously shown experiment with this strain (Figure 3C), the Aβ_1–42_ transgene was never induced, allowing aggregation and paralysis to develop gradually as worms age (see control curve in Figure 5). PNR886 and PNR962 extended the median times to paralysis by 30% and 58%, respectively (Figure 5), consistent with their amelioration of age-dependent aggregation.

### 2.3. PNR886 and PNR962 Bind the Inactive-GSK3β Allosteric Pocket More Avidly than TDZD-8

In a previous computational study, we used Glide docking and MM-GSBA analysis to predict that TDZD-8, a lead TDZD compound, would bind preferentially to the allosteric hydrophobic pocket of the GSK3β inactive conformation [39]. We repeated the same computational modeling, docking, and simulation procedures for binding of TDZD-8, PNR886, or PNR962 to the allosteric pocket unique to the inactive conformation of GSK3β. We first employed AutoDock-Vina to perform unbiased docking of each drug to inactive GSK3β, and, as observed previously [39], stable binding was predicted only within the allosteric pocket of GSK3β in its inactive conformation (Figure 6A–C, next page), which was then used in Glide for high-precision targeted docking. The GSK3β amino acids interacting with these drugs are listed in Table 1.

The implicit solvent-based binding free energies (ΔG_binding_) of PNR962 and PNR886, docked to the allosteric pocket of GSK3β, were estimated from MM-GBSA simulations (molecular mechanics of the generalized Born model with solvent accessibility) [51]. PNR962 and PNR886 are predicted to have 3–6 kcal/mol greater binding affinity than TDZD-8 (Figure 6D).

We then performed 200 ns atomistic GSK3β::ligand simulations in Desmond. After high-precision docking of each drug to the inactive conformation of GSK3β [39], we calculated the predicted binding free energy and, from that, the expected K_i_ for target binding. These data are summarized in Table 2 and Figure 7.

### 2.4. Ligand Free-Energy Perturbation Predicts Improved GSK3β Binding by New Analogs

Since PNR886 and PNR962 are analogs of TDZD-8, we used in silico ligand exchange to reassess their relative binding affinities for GSK3β. Free-energy perturbation, conducted within the Schrödinger suite, predicts the net energy shift due to an instantaneous ligand exchange. Briefly, after calculating the binding energy for a GSK3β complex with TDZD-8, that ligand was virtually converted to either PNR886 or PNR962, and the free-energy shift (ΔΔG) due to that replacement was calculated. Both PNR886 and PNR962 are predicted to have negative ΔΔG, indicating that each exchange from the lead molecule (TDZD-8) would enhance the stability of the complex (Figure 8). This procedure supports the results of MM-GBSA (Figure 6D); in both methods, PNR962 had the greatest affinity for GSK3β, but PNR886 was also superior to TDZD-8.

## 3. Discussion

Protein conformations are honed by natural selection to optimize biological functions. Many studies have demonstrated in vivo, in situ, and in silico that >90% of missense mutations altering the amino acid sequence are deleterious, chiefly because they tend to perturb normal protein folding [52,53,54]. Moreover, external factors such as environmental stresses (altered temperature or pH, or oxidative damage) can exacerbate this disruption, preventing a protein from attaining a functional or stable structure, subsequently contributing to age-progressive diseases [15,54,55,56,57]. Protein misfolding is held in check by chaperones that assist refolding and by clearance of misfolded proteins in proteasomes and autophagosomes. Proteasomes degrade polyubiquitin-tagged proteins, whereas autophagy removes larger structures including protein aggregates and damaged organelles [58].

A broad range of neurodegenerative diseases, including Alzheimer’s, Parkinson’s, and Huntington’s diseases, show distinct, diagnostic protein aggregates in or around neurons, which increase in number and size with disease progression and/or aging, which is itself the largest nongenetic risk factor for each disease [50,59]. A few non-steroidal anti-inflammatory drugs (NSAIDs), including aspirin and ibuprofen, as well as non-pharmacological interventions such as improved diet and exercise, have been proposed to confer protective benefits against aggregation-associated neurodegenerative diseases [18,60,61,62]. However, no small molecules have been shown to effectively halt or reverse progression of these diseases, despite considerable effort. Anti-inflammatory drugs have shown the most therapeutic promise against neurodegenerative diseases in prospective trials, which may be secondary to their anti-aggregation effects; they are also therapeutic against certain cancers, cardiovascular diseases, and type-II diabetes [1,42,60,63,64,65,66,67]. We synthesized and tested second-generation drugs of an established, potent family of NSAIDs, the TDZDs.

Our screens of drug candidates for neurodegenerative-disease amelioration or prevention have focused on inhibition of protein aggregation, reasoning that aggregate accrual is diagnostic of most or all such diseases and may provide a useful readout for the underlying mechanism. Interventions that alleviate protein aggregation are generally beneficial, often extending lifespan (i.e., reducing all-cause mortality). To date, curative or protective effects of NSAIDs in humans have been marginal, perhaps due to focusing on specific target enzymes (e.g., Cox-1 and Cox-2) rather than aggregation. Our aggregation-centered approach recently led to our identification of aggregation-inhibiting analogs of combretastatin-A4, and pursuit of PNR502 as a novel lead-drug candidate [42].

In the present work, we screened a structural library of TDZD analogs and identified two TDZD-related compounds, PNR886 and PNR962, which are predicted to surpass TDZD-8 as inhibitors of protein aggregation, and specifically of GSK3β. They may thus hold therapeutic potential to attenuate or reverse AD and other age-associated neuropathologies that respond to GSK3β inhibitors. These novel small molecules reduce the aggregate burden in a variety of aggregation-model systems, including two human cell-culture models of AD: SY5Y-APP_Sw_ neuroblastoma cells (which express APP_Sw_ and thus promote accumulation of extracellular amyloid) and HEK-tau embryonal kidney cells that have neuronal properties (expression of neurofilament proteins) and also express a tau transgene to form paired-helical filament inclusions (tau tangles). We quantified aggregates in cultured cells chiefly by staining with thioflavin T (Sigma-Aldrich), a molecular-rotor dye that shifts its fluorescence spectra when immobilized by binding to aggregates. Nearly identical results were obtained with Proteostat (Enzo Life Sciences), an aggregate-specific dye alleged to stain a broader range of aggregate types (Pearson correlation coefficient between these two dyes, over nine cultures, was 0.98 [*p* < 1 × 10^−20^]).

These drug candidates are also effective in whole-animal models, protecting *C. elegans* strains from extensive aggregate accrual due to their expression of aggregation-predisposing transgenes. It is intriguing that these novel TDZD analogs also extend *C. elegans* lifespan and healthspan, expanding on our previous findings of similar (but lesser) benefits conferred by other drugs that diminish protein aggregation [18,42]. It is encouraging that many drugs that reduce protein aggregation in cell models of AD also confer protection for behavioral traits in whole-animal models such as nematodes (*C. elegans*), insects (*D. melanogaster*), zebrafish (*D. rerio*), or mice (*M. musculus*) [63,68,69,70,71,72,73,74,75,76].

Numerous research reports have implicated GSK3β as a key kinase in AD, targeting and hyperphosphorylating tau and other proteins [77,78,79,80,81,82,83] believed to play causal roles in disease etiology. Overexpression of GSK3β is associated with tau hyperphosphorylation, reduced levels of nuclear β-catenin [84,85,86], and modification of other targets thought to be involved in AD pathology [84,87,88,89]. The proposal that GSK3β may play a pivotal role in AD etiology, termed the “GSK3 hypothesis of AD” [90], posits that pathogenesis of sporadic and familial types of AD may involve GSK3β hyperactivity leading to memory impairment, tau hyperphosphorylation, Aβ plaque accumulation, and development of plaque-related microglia-associated inflammation, and other inflammatory processes through activation of transcription factor NF-κB [91,92,93]. This hypothesis has led to considerable interest in GSK3β inhibitors as therapeutic agents for AD and other neuropathologies.

Since the mid-1990s, GSK3β has been considered as a therapeutic target for type 2 diabetes. That perspective arose from familial studies of insulin-resistant diabetes, which established that GSK3β is required for insulin signal transduction [94]. GSK3β phosphorylates, and thus inactivates, glycogen synthase (GS)—the enzyme that polymerizes glucose to form glycogen. Receptors for insulin and insulin-like growth factor (IGF-1) activate (via IRS1) class-I PI3 kinase (phosphatidylinositol-4, 5-bisphosphate 3-kinase), which plays a dual role in activating AKT; pAKT can then phosphorylate and inhibit GSK3β [95]. Thus, insulin signaling causes GS to produce glycogen, notably in the liver and muscle. Type 2 diabetes (T2D) is associated with hyperactivation of GSK3β, suppressing both glycogen synthase and insulin receptor substrate 1 (IRS-1) [94]. While T2D raises the risk of vascular dementia, it does not predispose to Alzheimer’s disease [96,97,98].

We note that GSK3β has numerous “off-target” substrates, the phosphorylation of which is neuropathogenic; these include tau, Aβ, α-synuclein, and TDP-43 [96,99]. The insulin-triggered kinase cascade, in its simplest form, may be represented as follows:
$$\begin{array}{ll} \mathrm{Ins}::\mathrm{Ins-R}\;\to \;\mathrm{IRS}1\;\to \;{\mathrm{PI}3K}_{I}\;\to \;({\mathrm{PIP}}_{3})\;\to \;\mathrm{AKT}1/\mathrm{AKT}2 & \;⊣\;\mathrm{GSK}3\beta \;⇥\;\mathrm{tau},\;A\beta ,\;\alpha \mathrm{-synuclein}\\ & ↘⊣\;\mathrm{GS}\;\to \mathrm{glycogen} \end{array}$$
where “→” and “⊣” indicate activating and inactivating phosphorylations, respectively, and “⇥” represents pathological phosphorylation favoring misfolding or aggregation of the target proteins. The pathways impinged by GSK3β are actually far more complex than this simplified version and include interactions with transcription factors, MAPK pathways, and AMPK and mTOR signaling, among others (e.g., see [95]).

It should be noted, however, that nothing in our study or in the literature establishes that GSK3β is the *only* target of these or other inhibitors. It remains possible that additional proteins, including other kinases, may also be TDZD-analog targets and may mediate or contribute to the observed protective effects of these molecules on protein aggregation, associated behavioral traits, and longevity or healthspan. In particular, we recently showed that interventions that slow the rate of translation also reduce aggregate burden [100]. Further studies (including manuscripts in preparation) address the identification and functions of other targets.

## 4. Materials and Methods

### 4.1. Effects of PNR886 and PNR962 on Protein Aggregation in Human Cells

Following previously described experimental procedures [101], two human cell lines were assessed for thioflavin T stainable aggregates: (*i*) human embryonic kidney cells that express normal tau (HEK293-tau) and form intracellular aggregates of processed tau fragments similar to those found in AD and AD-like-disease neurons; (*ii*) SH-SY5Y-APP_Sw_ neuroblastoma cells (expressing an aggregation-prone “Swedish” double mutant of amyloid precursor protein, APP, a generous gift from Dr. Mark Mattson, John Hopkins). Cells were cultured in DMEM supplemented with 10% (*v*/*v*) fetal bovine serum (FBS) at 37 °C, in Petri dishes. For subculture (replating), cells were detached by covering with trypsin/EDTA for 3–4 min at room temperature, after which medium was replaced and cells were rinsed 3 times in buffer before replating or harvesting. Prior to the assay, cells were grown for 48 h in the presence of varying doses (0.001, 0.01, 0.1 and 1 μM, as shown) of TDZD-8, PNR886, or PNR962 dissolved in DMSO (0.02% final DMSO concentration in culture medium) or the same final DMSO concentration for control cells. Treatment with PNR886 or PNR962 was either concurrent with replating or started 1 h prior to replating (pre-treatment) in order to ascertain for each drug the extent of protection conferred against protein aggregation.

### 4.2. Thioflavin T Staining of Cultured Human Cells to Quantify Relative Aggregation Levels

To assess the protective effects of the novel TDZD analogs on protein aggregation, HEK293-tau cells or SY5Y-APP_Sw_ neuroblastoma cells were grown in DMEM medium containing 10% fetal calf serum in T75 flasks, for 26 h at 37 °C (sufficient for a single doubling). After 48 h in the presence of several concentrations of PNR886 or PNR962, TDZD-8 or vehicle, cells were fixed in formaldehyde (4% *v*/*v*; 15 min. at 22 °C), washed, and stained 20 min in a dark container with 0.1% *w*/*v* thioflavin T mixed with DAPI (1 µg/mL; Life Technologies, Grand Island, NY, USA). After four washes in PBS, cells were covered with AntiFade (Life Technologies Inc.) and their fluorescence images captured on a Nikon DS-Fi2 camera mounted on a Nikon C2 inverted microscope with a motorized stage for automated well-by-well imaging, using appropriate filters: DAPI/blue (excitation 358 nm, emission 461 nm) and thioflavin T/green (excitation 385 nm, emission 450 nm). Immunohistochemical methods were as described previously [18]. The intensity of thioflavin T fluorescence per field was quantified via an ImageJ (1.54f) plug-in developed in-house and normalized to the number of DAPI-positive nuclei counted per field to obtain an average intensity of aggregates per cell for each treatment.

### 4.3. C. elegans Strains

All nematode strains used in this research were obtained from the Caenorhabditis Genetics Center (CGC; Minneapolis, MN, USA): wild-type Bristol-N2 [DRM lineage]; CL4176 [*smg-1*^ts^; *myo-3p*::*Aβ*_1–42_::*let-851* 3′-UTR; *rol-6(su1006)*], which expresses human Aβ_1–42_ in body wall muscle; and AM141 [*rmIs133; unc*-*54p::Q40::YFP*] expressing polyglutamine [Q40] fused in-frame to YFP [Q40::YFP] in muscle cells. *C*. *elegans* worms were maintained at 20 °C on 2% (*w*/*v*) agar plates in nematode growth medium (NGM), seeded in the center with *E. coli* strain OP50.

### 4.4. Effects of TDZD Analogs PNR886 and PNR962 on Aggregation in C. elegans Strain AM141

Fresh agar plates supporting bacterial lawns, which serve as food for *C*. *elegans*, were prepared at least one day ahead of use. *C. elegans* worms of strain AM141 were treated on these plates with each compound at final concentrations of 1–40 µM. Individual gravid adult worms were “axenized”, i.e., lysed in alkaline hypochlorite to recover a synchronous cohort of eggs. Drugs were allowed to diffuse throughout the medium on plates (~60 h), before adding eggs. Only the TDZD type and doses were varied; all other conditions were held constant as AM141 larvae hatched and developed into adults. In each experiment, a control group was treated with vehicle only (DMSO/medium) at the same final concentration, as a reference baseline for comparison to treated groups. *C. elegans* eggs at 20 °C normally hatch in ~8 h. Young-adult AM141 (3.5 days post-hatch) were placed on drug or control plates and transferred on alternate days onto fresh plates with the same drug doses, plus fresh *E. coli* to avoid even transient starvation. Equal samples (each *N* = 25) were randomly picked from experimental and control groups for microscopic imaging to quantify aggregates.

An in-house plug-in for ImageJ software (NIH) was used to process ~30 images of AM141 adults for each group, counting the number of aggregate foci per worm and measuring fluorescence intensity per aggregate (as an indication of their amyloid content), summarized in a spreadsheet. Each experiment was conducted multiple times to ensure reproducibility. Data were analyzed for significance of inter-group differences by heteroscedastic, two-tailed *t* tests, and displayed as histograms (mean ± SEM) of counts or intensity per worm for each treatment or control group.

### 4.5. Paralysis Assay in the Aβ-Transgenic Nematode Strain CL4176

To obtain synchronized cohorts of transgenic *C. elegans* strain CL4176, expressing Aβ_1–42_ in muscle after induction, worms were maintained at 20 °C on plates with ample *E. coli* (strain OP50). At day 3.5 post-hatch (after ~24 h as adults), *C. elegans* worms were lysed, and unlaid eggs were released and transferred to 100-mm Petri dishes containing NGM–agar seeded in a central area with OP50 bacteria. Plates were pre-equilibrated with PNR886 and PNR962 (at varying doses) or vehicle, each with a final concentration of 0.02% *v*/*v* DMSO. Expression of the human Aβ_1–42_ transgene was induced by upshifting *C. elegans* to 25.5 °C at the L3-L4 transition, with an assay after a further 48 h; alternatively, worms could be allowed to age without induction and assayed at a series of later times. Paralysis was assayed as described previously [50,102,103].

### 4.6. Lifespan Studies on the Nematode C. elegans Wild-Type Strain, Bristol-N2/DRM

To obtain synchronized cohorts for lifespan studies, Bristol-N2 worms (DRM culture) were maintained continuously for 3 generations, avoiding contamination and starvation. Healthy, well-fed adult worms from the third generation were then lysed and eggs placed on plates equilibrated with appropriate doses of PNR886, PNR962, or DMSO vehicle (each at a final concentration of 0.02% *v*/*v* DMSO). Beginning at the L4 larval stage, worms were transferred daily to fresh plates for 7 days and thereafter on alternate days until the last worm died. *C. elegans* worms that moved spontaneously or responded to gentle prodding were scored as alive. Worms lost for reasons other than natural death were censored after their last observation reported as “alive”. The methods described above are adapted from previously published experimental procedures [104,105].

### 4.7. Structural Modeling of TDZD Analogs and GSK3β

We used the inactive conformation of GSK3β from our previous work [39], converted to Autodock format. Structures of PNR886 and PNR962 were converted to the simplified molecular-input line-entry system (SMILES) format using ChemDraw and subsequently converted to the SYBYL mol2 format. SMILES uses ASCII strings for line notation to represent the structures of chemical species. Because the published structure of the inactive-conformation protein is incomplete, we filled in missing hydrogens, side chains, and omitted loops by template modeling using the in-built protein preparation wizard from Maestro Prime module (Schrödinger, Inc., New York City, NY, USA). All subsequent computational-docking and simulation studies used the preprocessed inactive conformation of the GSK3β protein template as described [39].

### 4.8. Docking of TDZD Analogs to the Inactive Conformation of GSK3β

Computational modeling, docking, and simulation methods follow our established protocol described previously [39]. Unbiased or blind docking was conducted with Autodock-Vina using the Raccoon interface via a Linux-based server. Unbiased docking entails enclosure of the full-length protein in a grid box, thereby allowing the ligands (PNR886 and PNR962) to search for their optimal binding sites on the entire target protein (GSK3β in inactive conformation) as previously described [39]. This was followed by targeted docking with the Glide module and MM-GBSA plug-ins from Schrödinger suite, applied to the full-length inactive conformation of GSK3β that we previously evaluated [39]. For Glide, a grid box was generated using the Receptor Grid Generator Wizard in Schrödinger Maestro; docking was then performed in standard-precision mode. Visualization and analysis of results were conducted in Maestro Viewer and Discovery Studio Visualizer. The predicted drug-binding sites, as described, provided targets for Glide docking, interfaced with Maestro 2017-2 Suite (Schrödinger), conducting MM-GBSA analysis to evaluate stability of fully solvated protein–ligand targeted docking [39].

### 4.9. Atomistic Molecular-Dynamic Simulations

For protein–ligand simulations, each ligand-protein complex (PNR962::GSK3β, PNR886::GSK3β and TDZD-8::GSK3β) was enclosed in an orthorhombic box with edges passing within 10 Å of the protein. Solvation and charge neutralization within the box were accomplished with simple point charge (SPC) water and Na^+^, Cl^−^ counterions, respectively, with a final physiological NaCl concentration of 0.15 M. The system was equilibrated at 300°K using the Nosé–Hoover chain thermostat and NVT protocols for simulation. After a secondary equilibration under NPT conditions, the Molecular Dynamic (MD) simulation was run as described previously [106,107], holding temperature (T) at 300 °K and pressure (P) at 1.0 bar with the RESPA integrator. All simulations were conducted in Desmond v2018.1 on an enhanced in-house cluster deploying NVIDIA Quadro P5000 GPUs. Simulation trajectories for protein–ligand complexes were viewed and analyzed using the Simulation Interaction Diagram Generator module in Desmond-Maestro to assess protein–ligand interaction stability and ligand-induced protein structural changes.

### 4.10. Binding Energy (ΔG_binding_) Computation in MM-GBSA

Glide docking poses served as the initial inputs for computing the retrospective solvent-based free energies (ΔG_binding_) of ligand-protein complexes (i.e., PNR962::GSK3β, PNR962::GSK3β, and TDZD::GSK3β). To estimate the binding free energy of individual ligands attached to GSK3β, we employed the inbuilt Prime module of Schrödinger Suite for the MM-GBSA procedure. Ligand exchange to compare GSK3β binding to TDZD-8 vs. PNR886 or PNR962 provides reliable estimates of ligand-free-energy changes. Using results from Glide docking described above, the inactive conformation of the GSK3β target protein is complexed with PNR886 and PNR962 and exported in the Schrödinger Maestro 2017-2 Suite. The binding energy of TDZD-8 to GSK3β was computed (represented as ΔG_binding_) and then TDZD-8 is replaced by PNR886 or PNR962, after which the change in binding free energy is computed using the Ligand-FEP protocol from Schrödinger Suite. The difference between these energies, denoted as ΔΔG of binding to GSK3β (kcal/mol), represents the ligand free energy change due to the perturbation (ligand exchange).

### 4.11. Statistical Analyses

Significances of survival-curve differences were assessed pairwise by Gehan–Wilcoxon log-rank tests. Between-group differences in measures of protein aggregation, chemotaxis, and paralysis experiments, treating replicate assays as individual points, were assessed by Fisher–Behrens heteroscedastic *t* tests appropriate to samples of unequal size or unknown variance (usually due to low *N*). Two-tailed *t* tests were used if the direction of change was unknown but were replaced by one-tailed tests once that direction was established. To assess results from staining with thioflavin T or antibody, the mean counts or fluorescence intensities of 15 random fields per well were treated as individual points for each treatment group. Chi-squared or Fisher exact tests (depending on N per group) were used to evaluate inter-group differences in proportions (the fraction of paralyzed or chemotactic worms) within individual experiments.

## 5. Conclusions

Protein aggregation and aggregation-associated behavioral traits seen in neurodegenerative pathological models are strikingly inhibited by unique TDZD analogs, PNR886 and PNR962. These novel drugs ameliorated protein aggregation in both human neuronal and embryonic kidney cell models and in nematode *C. elegans* models of neurodegeneration-associated aggregation. Future research will employ in vitro and in vivo approaches to elucidate whether protein targets other than GSK3β are involved in the benefits conferred by PNR886 and PNR962. Further characterization and mechanistic studies on these novel analogs and their targets might offer novel and effective pharmacological interventions for diverse neurodegenerative diseases —Alzheimer’s and Parkinson’s diseases, amyotrophic lateral sclerosis, etc. Hyperphosphorylation of “off-target” protein targets, by GSK3β and other kinases, contributes to protein misfolding and aggregation that are strongly associated with a wide variety of neurodegenerative diseases.

## Figures and Tables

**Figure 1 pharmaceuticals-16-01498-f001:**
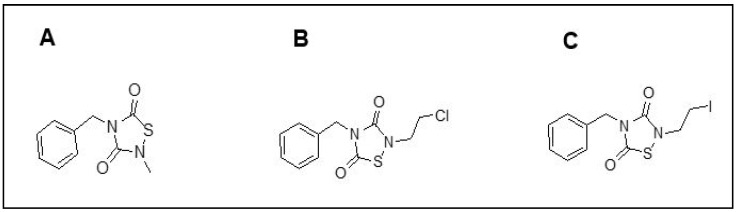
**Structures of TDZD-8 and two novel anti-inflammatory drugs, PNR886 and PNR962.** TDZD analogs were synthesized with structures shown: (**A**) TDZD-8; (**B**) PNR886; and (**C**) PNR962.

**Figure 2 pharmaceuticals-16-01498-f002:**
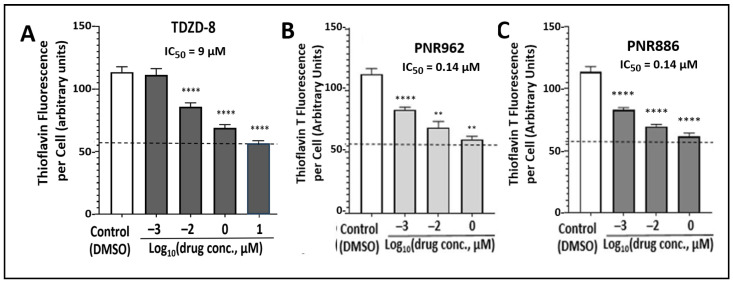
**TDZD analogs PNR886 and PNR962 are more potent than TDZD-8 in reducing HEK-tau aggregation**. HEK293-tau cells were cultured 48 h in the presence of the indicated doses of TDZD-8 (**A**), PNR962 (**B**), or PNR886 (**C**), displayed on a log(10) scale, comprising zero (controls), 1 nM, 10 nM, 1 µM, and (for TDZD-8) 10 µM. Cells were then stained with thioflavin T, and aggregate signal per cell was quantified from fluorescence images (see Materials and Methods). Significance of differences from controls (combined from two to four repeats) was assessed by two-tailed *t* tests, with nominal adjustment of the alpha level (α < 0.01) to compensate for multiple comparisons. ** *p* < 0.01, **** *p* < 0.0001. Based on fluorescence per cell at 1 nM (10^−3^ µM) of each drug, both PNR962 and PNR886 were significantly more effective than TDZD-8 at reducing tau-mediated aggregation (*p* < 10^−4^).

**Figure 3 pharmaceuticals-16-01498-f003:**
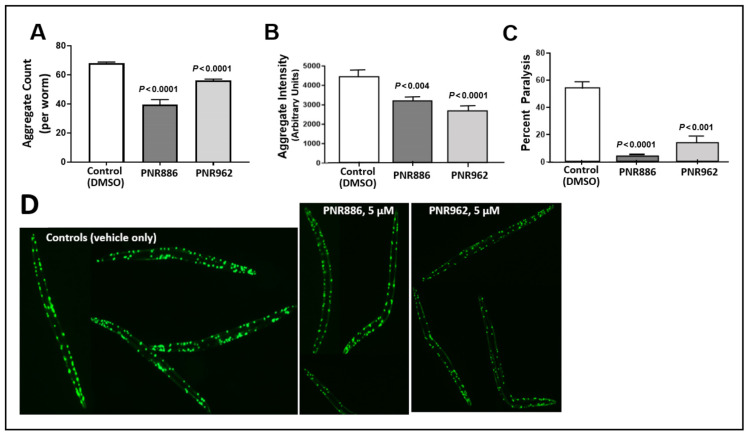
**TDZD analogues reduce aggregation and paralysis in *C. elegans* models of HD.** (**A**,**B**) Adult worms (*C. elegans* strain AM141, expressing Q40::YFP) were exposed to PNR886 or PNR962 from the end of larval development (late L4) through adult day 3, and imaged on adult day 3. Images were processed to quantify (**A**) the number of aggregate foci and (**B**) total worm fluorescence of foci as a measure of amyloid/worm. (**C**) The percent of paralyzed worms is shown for *C. elegans* strain CL4176, 48 h after inducing Aβ_1–42_ expression in muscle by upshift to 25 °C at the L3/L4 transition. Worms were exposed to PNR886 or PNR962 from larval L3/L4 stages through day 3.5 post-hatch, i.e., 2 days prior to image capture. Significance (as shown) was assessed by two-tailed heteroscedastic *t* tests, with adjustment of the alpha level (to α = 0.01) to compensate for multiple comparisons. (**D**) Typical AM141 images.

**Figure 4 pharmaceuticals-16-01498-f004:**
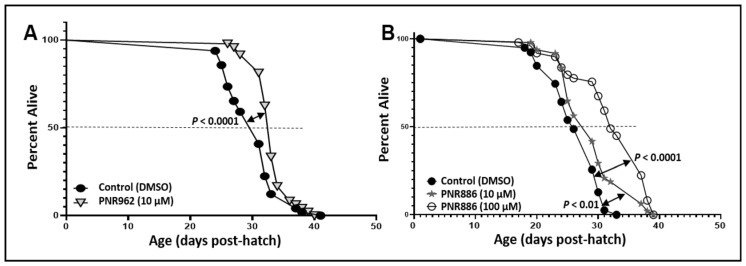
**PNR886 and PNR962 extend lifespan of wild-type *C. elegans*.** Lifespan survivals are shown for wild-type *C. elegans* (strain Bristol-N2, DRM lineage) exposed to (**A**) 10 µM PNR962 or (**B**) 10 or 100 µM PNR886, continuously from the L4/adult molt, replenished on fresh plates every other day. In each panel, drug treatments are compared to vehicle alone (controls) containing DMSO at the same final concentration as utilized in the drug plates. Worms (two plates of 25 worms per group, *N* ≈ 50) were maintained at 20 °C with ample *E. coli* (strain OP50) as food. Significance of survival differences, drug-treated vs. controls, were assessed by Gehan–Wilcoxon log-rank tests.

**Figure 5 pharmaceuticals-16-01498-f005:**
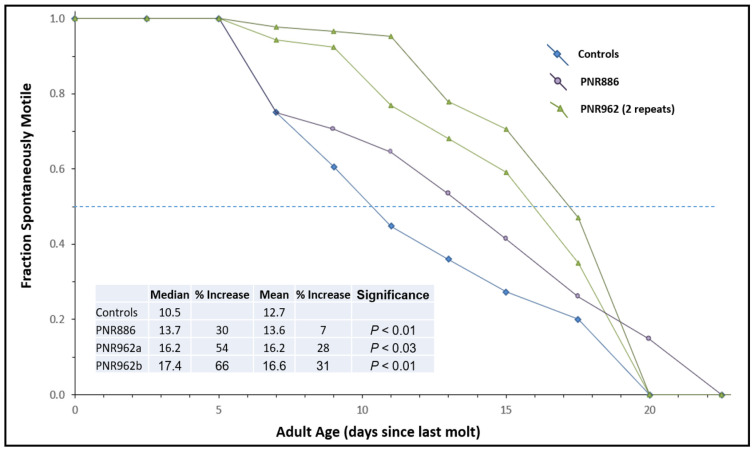
**Age-dependent loss of motility is reduced by PNR886 and PNR962.** TDZD analogs were added to NGM–agar plates at 10 µM, and 50 L4 larvae (*C. elegans* strain CL4176) were placed on these plates and transferred to fresh plates with drugs every 3 days. Cohorts were analyzed as for lifespan survivals, except immobile and dead worms, which were combined to calculate the paralyzed fractions. Gehan–Breslow–Wilcoxon tests were used to assess significance of differences from controls to allow disproportional hazards.

**Figure 6 pharmaceuticals-16-01498-f006:**
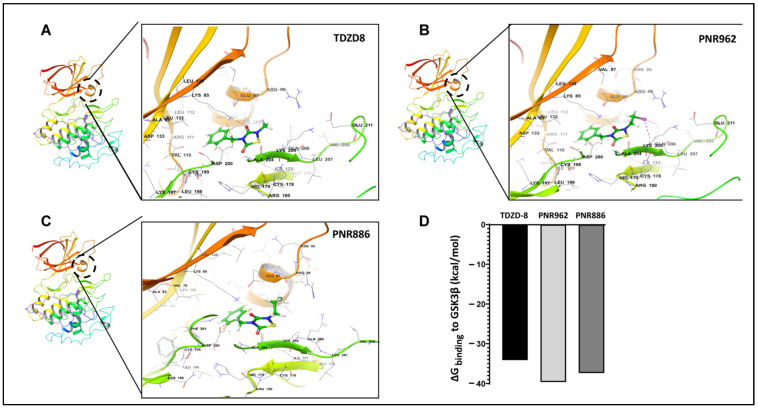
**TDZD analogues PNR886 and PNR962 bind the hydrophobic pocket of GSK3β in the inactive conformation**. PNR962, PNR886, and TDZD-8 bind to the allosteric hydrophobic pocket of GSK3β. (**A**) Glide docking pose of TDZD-8 binding to the GSK3β allosteric pocket in the modeled inactive DFG-out conformation. (**B**) Glide docking of PNR962 binding to the GSK3β allosteric hydrophobic pocket in the modeled inactive DFG-out conformation. (**C**) Glide docking of PNR886 binding to the GSK3β allosteric hydrophobic pocket in the modeled inactive DFG-out conformation. (**D**) MM-GBSA-based ΔG_binding_ (Gibbs free energy of binding) calculated for TDZD analogues (TDZD-8, PNR962, and PNR886), each with GSK3β in the inactive (DFG-out) conformations. Molecular structures are depicted with Schrödinger Maestro 11.4 (Schrödinger Inc., New York City, NY, USA).

**Figure 7 pharmaceuticals-16-01498-f007:**
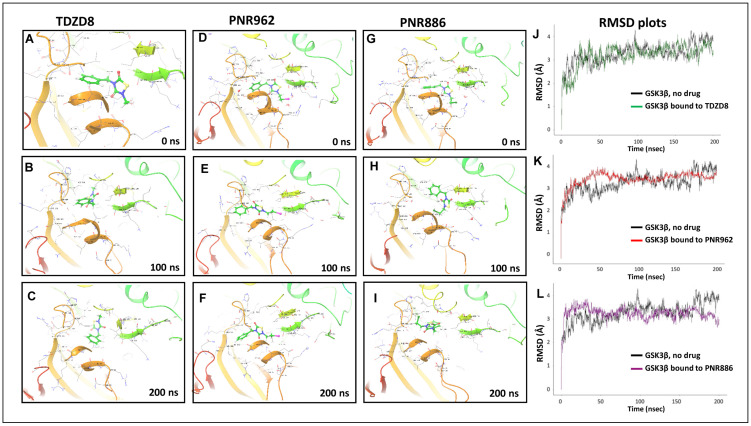
Molecular-Dynamic simulations support stable binding of TDZD analogs TDZD-8, PNR886, and PNR962 to the hydrophobic pocket in inactive GSK3β. (**A**–**I**), Snapshots from 0.5-µs simulations of full-length GSK3β in the inactive conformation, bound to TDZD-8, PNR886, or PNR962; structural images were extracted at 0 ns, 100 ns, and 200 ns. (**J**–**L**), Root Mean Square Deviations (RMSD) time-courses of GSK3β::ligand complexes are plotted for simulations of GSK3β bound to TDZD-8, PNR886, or PNR962, each superimposed over the same no-ligand control simulation (black tracings). Replicate simulations produced similar results, reaching plateaus (indicating stable conformations) in <50 ns. Simulations were conducted in Schrödinger Maestro 11.4 (New York City, NY, USA).

**Figure 8 pharmaceuticals-16-01498-f008:**
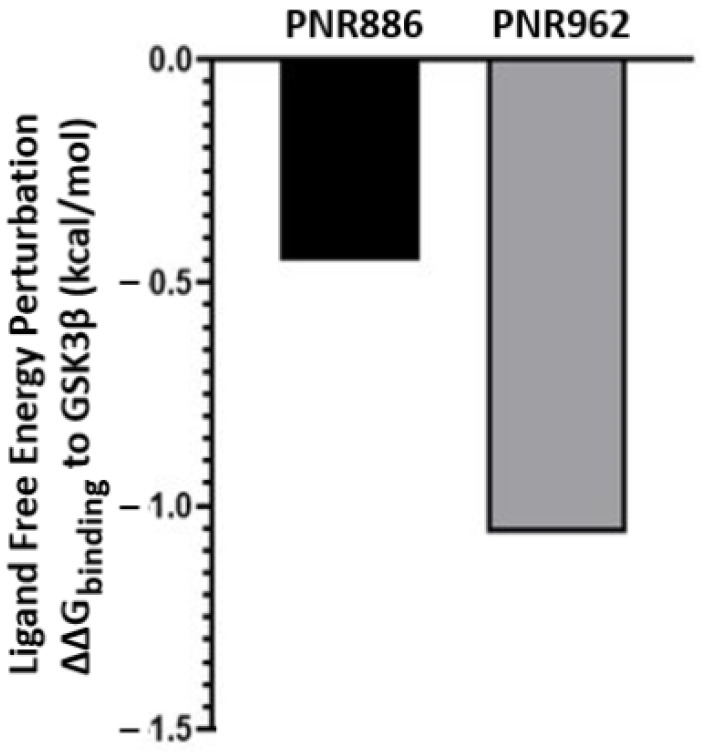
**Replacing TDZD-8 with PNR886 or PNR962 is predicted to improve binding to the hydrophobic pocket in the inactive conformation of GSK3β**. The binding energy of TDZD-8 to GSK3β is first computed (represented as ΔG), and then TDZD-8 is exchanged with PNR886 or PNR962 using the FEP+ module of the Schrödinger Suite, permitting calculation of the resultant shifts in free energy (denoted as ΔΔG for binding to GSK3β, in kcal/mol).

**Table 1 pharmaceuticals-16-01498-t001:** GSK3β amino acids interacting with TDZD analogs.

TDZD Analog	GSK3β Amino Acids Interacting with TDZD Analogs
TDZD-8	ILE100, MET101, VAL110, ARG111, LEU112, LEU132, ASP200, ALA204, LYS205, GLN206
PNR886	LYS85, GLU97, GLN99, ILE100, MET101, ASP200, PHE201, ALA204, LYS205, GLN206
PNR962	LYS85, ARG96, GLU97, ILE100, MET101, VAL110, ARG111, LEU112, LEU132, ASP200, ALA204, LYS205, GLN206
TDZD-8 & PNR886	ILE100, MET101, ASP200, ALA204, LYS205, GLN206
TDZD-8 & PNR962	ILE100, MET101, VAL110, ARG111, LEU112, LEU132, ASP200, ALA204, LYS205, GLN206
All Three Analogs	ILE100, MET101, ASP200, ALA204, LYS205, GLN206

**Table 2 pharmaceuticals-16-01498-t002:** **Predicted ΔG_binding_ and K_i_ of TDZD analogs**.

TDZD Drug	ΔG_binding_ (kcal/mol)	K_i_ (µM)
PNR886	−6.2	2.7
PNR962	−6.2	2.7
TDZD-8	−6.1	3.8

## Data Availability

Detailed data will be provided upon request, unless prevented by intellectual property considerations.

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
