# Peer review of "Thiadiazolidinone (TDZD) Analogs Inhibit Aggregation-Mediated Pathology in Diverse Neurodegeneration Models, and Extend C. elegans Life- and Healthspan"

_pharmaceuticals, 2023, doi:10.3390/ph16101498_

Round 1

Reviewer 1 Report

Kakraba et al. studied the action of Thiadiazolidinone derivatives against aggregation-mediated pathology in neurodegeneration models.  The authors investigated the inhibitory activity of synthesized compounds against aggregation of tau and Aβ1–42.

I have some major concerns which need to be addressed.

1.      Authors should calculate the IC50 value for these compounds against interested protein/peptide aggregation.

2.      Direct binding with respective protein/peptide (tau and Aβ1–42) should be performed using in-vitro assay and calculate  IC50.

3.

      Authors should perform a TEM study to verify the Thioflavin-T assay.

4.      Cytotoxicity study should be tested at higher concentrations, like 10-50 µM

5.      Authors should report the Ki value with binding energy in a molecular docking study. 

 Minor editing of English language required

Author Response

Thank you for your comments and suggestions. Please, find attached to this our responses to your comments and requests.  We left our comments in blue for easy identification and under each question. Thank you.

Reviewer 2 Report

The research article entitled “Thiadiazolidinone (TDZD) analogs inhibit aggregation-mediated pathology in diverse neurodegeneration models, and extend C. elegans life- and health-span” presented a nice work combining computational biology and experimental approaches, to identify potent thiadiazolidinones analogs that could inhibit amyloid-aggregation associated with neurodegeneration. The concept and execution of research is clear and impressive. In my opinion, the article could be accepted after addition of minor changes as stated below.

Kindly address the following comments:

1.      Minor typographical and grammatical errors could be observed in the present research article. I suggest rectifying all the typo and grammatical error from the article.

2.      Keywords must be in alphabetical order.

3.      In figure 6 and 7, it is very difficult to observe the interacting amino acids. I knew that it is very difficult to accumulate them in one figure. Thus, a table with the information about interacting amino acids would supplement it and help the readers to understand. I recommend adding a table of interacting amino acids with each ligand with discussion on the importance of these interactions with the amino acids (in the discussion section).

4.      Thiadiazolidinone multi-targeting ability against neurological disorders need to discussed with references. In fact, there are so many articles that showed the potency of anti-diabetic drugs against Alzheimer’s and other neurological disorders. Kindly add a paragraph with proper justification on the abovementioned topic.

5.      Future plan based on the results obtained need to be more elaborative in the discussion section. Kindly add few more sentences pertinent to it.

6.      Author’s opinion based on their expertise in the field at the end of conclusion is missing. I suggest authors should add their opinion to clarify their aspects on the topic.

1.      Minor typographical and grammatical errors could be observed in the present research article. I suggest rectifying all the typo and grammatical error from the article.

Author Response

Thank you for your suggested edits and review. We have attached our responses in the word file attached to this submission. Thank you.

Author Response

Thank you for your review and comments. We have responded to the questions and attached our responses to this submission. Thank you.

Round 2

Reviewer 1 Report

The authors corrected the ms as suggested. Now ms may be accepted for publication.

Reviewer 3 Report

I recommend that the present manuscript can be accepted for publication.